# Exploring the Relationship between the Clustering Degree of Children's Business Formats and the Attractiveness of Commercial Centers in Wuhan by Modifying the Classic Retail Model

**Zhuoran Shan, Xuehan Shen and Man Yuan ***

School of Architecture and Urban Planning, Huazhong University of Science and Technology,
The Key Laboratory of Urban Simulation for Ministry of Natural Resources, Hubei Engineering and Technology
Research Center of Urbanization, Wuhan 430074, China; 2017010012@hust.edu.cn (Z.S.);
M202173966@hust.edu.cn (X.S.)
*   **Correspondence: yuanman_aup@hust.edu.cn**

**Abstract:** In recent years, the continued popularity of children's consumption has made it a new factor that affects the attractiveness of Wuhan's commercial centers. This study analyzes the characteristics of children's business format agglomeration in Wuhan commercial centers based on the results of an on-the-spot investigation and estimates the passenger attraction of 66 commercial centers in the main urban area with the support of LBS data. In addition, we set up a control experiment group of commercial centers of various levels and explore the influence mechanism of the density of various types of children's stores on the attraction of commercial centers by improving the classic retail model. The results indicate the following. (1) Children's business formats in Wuhan's commercial centers are active as a whole, and different types of children's businesses have an unbalanced layout at the different levels of business centers. (2) There are both level ladder and internal level differences in the attractiveness of Wuhan commercial centers. (3) The direction and intensity of the influence of children's business types on commercial centers of different levels differ. In city-level commercial centers, children's education and entertainment formats play a role in promotion. In county-level commercial centers, the children's education format is the most important, and overinvestment in the children's department store format may not meet expectations. In community-level commercial centers, investment in children's department stores yielded the best results. (4) Traffic impedance has a stable inhibitory effect at all levels of Wuhan commercial centers, which is in line with the classic retail gravity theory. Further, based on the above results, this paper puts forward suggestions on several types of adaptations that can be applied to children's consumption stores at different levels of commercial centers to provide support for rationally utilizing the potential of the children's consumption market.

**Keywords:** children's business formats; commercial center; attractiveness; influence mechanism; retail model

## 1. Introduction

The attractiveness of commercial centers refers to their ability to attract customers from areas with concentrated commercial activities within a certain range, which is of great value to urban economic recovery in the post-epidemic era. Under the guidance of the common goal of promoting economic growth and social prosperity, the attractiveness of commercial centers has also become a focus of attention in the field of urban economy and planning of major cities worldwide [1–4]. In research on the factors influencing the attractiveness of commercial centers, numerous scholars find that commercial scale (usually expressed by the floor area of commercial centers) and traffic resistance have a steady impact on

the attractiveness of commercial centers [5–8], and the relevant planning practices of commercial centers have mature optimization technologies in these two aspects.

The clustering degree of children's business formats refers to the concentration of children's business formats within a certain area, which has become an important starting point for commercial operations in the period of expanding the children's consumer market. Growth in the number of consumers and improvement in consumption levels are key manifestations of the expansion of the consumer scale. Children play an increasingly active role in consumer groups [9], and the commercialization of children's space has also become a new development trend on a global scale [10]. Dotson et al. believed that the rapid growth of children's commercial space in the United States is an important factor in promoting the socialization of children's consumption [11]. Broberg et al. pointed out in their research on the structure of Finnish child-friendly cities that commercial spaces, such as shops, restaurants, and playgrounds, offer important environmental support for the construction of child-friendly cities [12]. Crewe et al. found that extensive children's retail commercial space is key to the success of the British children's fashion market [13]. According to Vinken's survey, the Japanese consumer market is generally connected to maternal and child services and children's education [14]. With the upgrading of consumer markets and the improvement of maternity security policies worldwide, it is evident that the "children's economy" is heating up. The children's market is characterized by the expansion of consumer scale and higher consumption levels. China is one of the most representative countries in the world. As of the end of 2020, children aged 0–14 accounted for 17.95% of the total population in China, an increase of 1.35% from A decade ago [15]. According to the "2018 China Children's Family Survey White Paper", the scale of China's children's consumption market has approached RMB 4.5 trillion, and the average annual expenditure per child can reach RMB 20,000. At the same time, marketers have become aware of children's influence on consumption, leading to a significant increase in the proportion of children's business formats in commercial centers. Among the top 100 newly opened shopping centers in China in terms of comprehensive strength, parent–child business accounts for 9.54%, making it the second-largest experience business after the catering business.

At present, both commercial consumption demand and the children's consumption market in Wuhan are expanding rapidly. Wuhan is an important commercial town that has "prospered by business" [16]. In the past ten years, the area of Wuhan's commercial center has increased significantly. In 2020, the area of Wuhan's commercial center will reach 9.45 million square meters, and the per capita commercial center area will reach 0.766 square meters, second only to Shanghai and Hangzhou. Moreover, the number of children with household registrations in Wuhan reached 1.478 million in 2020, accounting for 16.7% of the city's population. The overall population of children is growing and the demand for children's consumption is expected to further increase in the future.

Although commercial centers oriented to children's consumption have been proven to be a highly profitable business model in extensive commercial practice, it is common to see that the "high area" of children's business formats has not brought "high popularity" and "high income" in recent years. Such centers are not a band-aid for alleviating the impact of COVID-19 nor a panacea for offline consumption transformation due to their low rental capacity, large footprints, and low efficiency. Specifically, "how to play a positive role in children's formats while promoting the appeal of commercial centers" and "what are the differences in the influences of different types of children's formats on different levels of commercial centers" are urgent questions to answer.

In this paper, we first review the related research progress on children's business formats and the attractiveness of commercial centers. Second, we expound on the data and methods used for analyzing clustering degree and attractiveness before describing the features of children's business format agglomeration and commercial center attraction in Wuhan. Third, we revealed the influence mechanism of different types of children's business formats on the attractiveness of different levels of commercial centers. Finally, we

put forward suggestions on the adaptation, transformation, and developmental direction of children's business formats at various levels of commercial centers.

## 2. Literature Review

### 2.1. Research Progress on Children's Business in Commercial Centers

Children constitute a special group of consumers. Importantly, children's consumption behavior in commercial centers cannot simply be understood as parental subrogation consumption. Ironico asserted that children are not consumers in the full sense, but rather "progressing consumers" who constantly collect information and improve their skills in their consumption behavior. He also highlighted that children have a positive role in redefining commodities, stimulating commercial vitality, and encouraging the reuse of commercial spaces through play [17]. Berti et al. believed that children are not fully aware of the commercial nature of commodities, and preschool children usually use their intuition to explain purchase behavior and deepen their understanding through further learning in the consumption process [18]. McNeal argued that children can learn consumer behavior and meet creative activity needs by interacting with commercial products and spaces, even without the involvement of other social actors [19]. The influence of children's groups on family shopping decisions is significant and dynamic [11,20]. Miller et al. found that parents are not as rational in their consumption process as they think they are [21]. Hamilton et al. thought that parents would increase spending out of love and protect their children [22]. Several studies have shown that the influence of children on the family consumption process is difficult to assess [23–25].

As the influence of children's consumption gradually expands, education, entertainment, retail, and other types of children's business formats have increasingly become the mainstream configuration of commercial centers. Marketers have conducted extensive business practices centered on children's business formats [25]. This is especially true since children today spend more time in commercial settings, e.g., shopping centers, than ever before and are surrounded by commercial messages in a variety of forms [26]. On the one hand, diverse business models for children have emerged. Feenstra et al. pointed out that to build family traffic and develop consumption in the store, the business model of educational seminars and events for children is widely used, such as workshops, birthday services, and commercial events with national brands [27]. Wiener found that retailers adopt the "playtail" business model and organize activities to attract children to play with products, which can effectively enhance a store's attractiveness [28]. Guichard and Damay suggested that commercial events and workshops may fulfill some children's emotional and social expectations of an ideal store [29]. On the other hand, various children's business formats have gradually become an important part of urban commercial centers. The China Children's Business Formats Management Center divides children's business formats into four categories: children's education, retail, services, and entertainment. A research report on China's children's business format reveals that the top three brands reflect children's retail, education, and entertainment in terms of the number of business formats. Many scholars have provided insights into the role of various types of children's business formats in commercial centers. Zhang proposed that children's entertainment format is an essential resource for high-level commercial centers in Chinese cities [30]. Shen et al. argued that children's entertainment format is an indispensable part of today's commercial complexes in large cities in China [31]. Zhang et al. [32], Cai et al. [33], and Liu et al. [34] proposed increasing the proportion of children's education and entertainment formats to form essential differentiated competitive advantages with online e-commerce and to stimulate the vitality of physical business centers. Chan identified that the children's retail format is popular with urban children in China [35].

### 2.2. Related Research on the Attractiveness of Commercial Centers

Measurement methods and factors influencing the appeal of commercial centers have attracted the attention of many scholars for a long time. In terms of attracting customers, the

famous models include J. Reilly's "law of retail attraction", P.D. Converse's breaking point formula, A.G. Wilson's "entropy maximization model", D.L. Huff's probability model, G. Rushton's gravitational model, and R.M. Downs' shopping mall cognitive structure. The wide application of urban big data, such as GPS floating vehicle trajectory data and LBS location information service data, has progressively perfected the method to measure passenger attraction [36–38]. In terms of influencing factors, it has become a common understanding that the relationship between business center attraction and commercial scale is positively correlated, and the negative correlation with traffic impedance has been repeatedly tested by many scholars [5–8]. In recent years, research has extensively analyzed the influence of business formats, levels, consumer characteristics, development strategies, and the surrounding environment of commercial centers on customer attraction. Wang et al. separated the attractiveness intensity of commercial centers from the attractiveness range and proposed that commercial centers of all levels will have a trend of either "improvement" or "recession" under certain conditions [39]. Hao et al. revealed that different commercial centers show different levels and scales, and the scales and patterns of the spatial agglomeration of different formats also differ [40]. Wang et al. conducted an empirical study on customer attraction to community-level commercial centers in Xi'an and stated that it was related to the density of the street network and the number of schools [41]. Zhou et al. confirmed that the traditional geospatial attenuation law is disturbed locally owing to the influence of market rules, consumer preferences, and other factors. To a certain extent, the complementarity of business formats promotes interdependence among commercial centers [42]. Qin et al. observed that the spatial distribution of the catering format in Nanjing follows the central place theory, and the overall popularity is low [43]. Chen et al. found that the entertainment format is an essential factor in improving the attraction of commercial center passenger flows [44]. Lin et al. also believed that leisure and entertainment were the main purposes of physical consumption [45].

### 2.3. Construction and Optimization of the Retail Model

The retail model, favored by researchers for nearly half a century, is based on Newton's law of gravitation and applicable to the retail field. With ongoing improvements in research, it has matured and is now widely used in studies on the passenger attraction mechanism of urban commercial centers. In the 1930s, Reilly first introduced Newton's gravitational model to the study of urban economic geography when analyzing the retail relationship in Texas, demonstrating that the commercial attraction of a city is directly proportional to the size of the urban population and inversely proportional to the square of the distance [46]. Hotelling further pointed out that consumer payment of transportation costs is key to price competition in commercial centers [47]. Subsequently, other consumer behavior research revealed that consumers exhibit a wide range of "irrational" economic and spatial behaviors [48], thus challenging the single commercial center minimum effort rule in traditional models [49]. Huff confirmed that consumers' purchasing behavior exhibits spatial randomness and proposed a Huff model to measure the interaction probability between consumers and retail stores in a specific location [50]. Lashmanan and Hansen constructed a shopping model closely related to the transportation model, based on network tracking and the gravity model trip distribution theory [51]. Wilson introduced the concept of "entropy" to the study of social geography and described the dynamic equilibrium system constructed by the retail gravity model with the process of entropy maximization [52]. The model was later applied on a smaller scale and extended by Fotheringham and Knudsen to study location rents, external economies of scale, and retail agglomeration [53]. Dearden and Wilson further refined the model, identifying discontinuities in the Greater London retail system and arguing that understanding the underlying discontinuities of a retail system can provide effective advice to policymakers and retail businesses [54]. In 2018, Wilson further optimized the dynamic retail model and introduced a system of stochastic differential equations (SDEs) to solve fluctuations that the system may cause due to force majeure factors. The construction of this model has remained relatively stable for decades [55].

Since then, researchers have improved the retail gravity model with multiple disciplines and proposed numerous application methods. For example, Piovani et al. combined the gravity model with percolation theory to explain the impact of transportation networks on the attractiveness of commercial centers [56]; Schlaich et al. used a retail gravity model to simulate consumer shopping interactions related to the retail food environment, nutrition, and public health [57]; Ellam et al. simulated and explored the dynamic characteristics of the spatial interaction of the UK retail system by constructing a stochastic differential equation [58]. Shan et al. explored the influence mechanism of the consumer age, industry type, and development time on the attractiveness of the Wuhan commercial center by improving the classic retail attraction model [36].

### 2.4. Insufficient Existing Research

The existing studies have three shortcomings: (1) less attention has been paid to the agglomeration characteristics of children's business formats, and more empirical research is needed to support them; (2) although the promotion of children's business formats on attractiveness has been widely recognized, how different types of children's business formats contribute to the attractiveness of commercial centers at different levels remains unclear; and (3) the classic retail gravity model does not include the children's business format factor; thus, it needs to be improved and reconstructed to meet the current development needs. To address these shortcomings, this study used modeling ideas from social physics and statistical mechanics to improve the classic retail gravity model. To that end, it includes variables of the clustering degree of children's business formats (i.e., children's retail, entertainment, education, and other formats). Furthermore, a commercial center-level control group experiment was conducted to test the variation in the attractiveness mechanism (Figure 1).

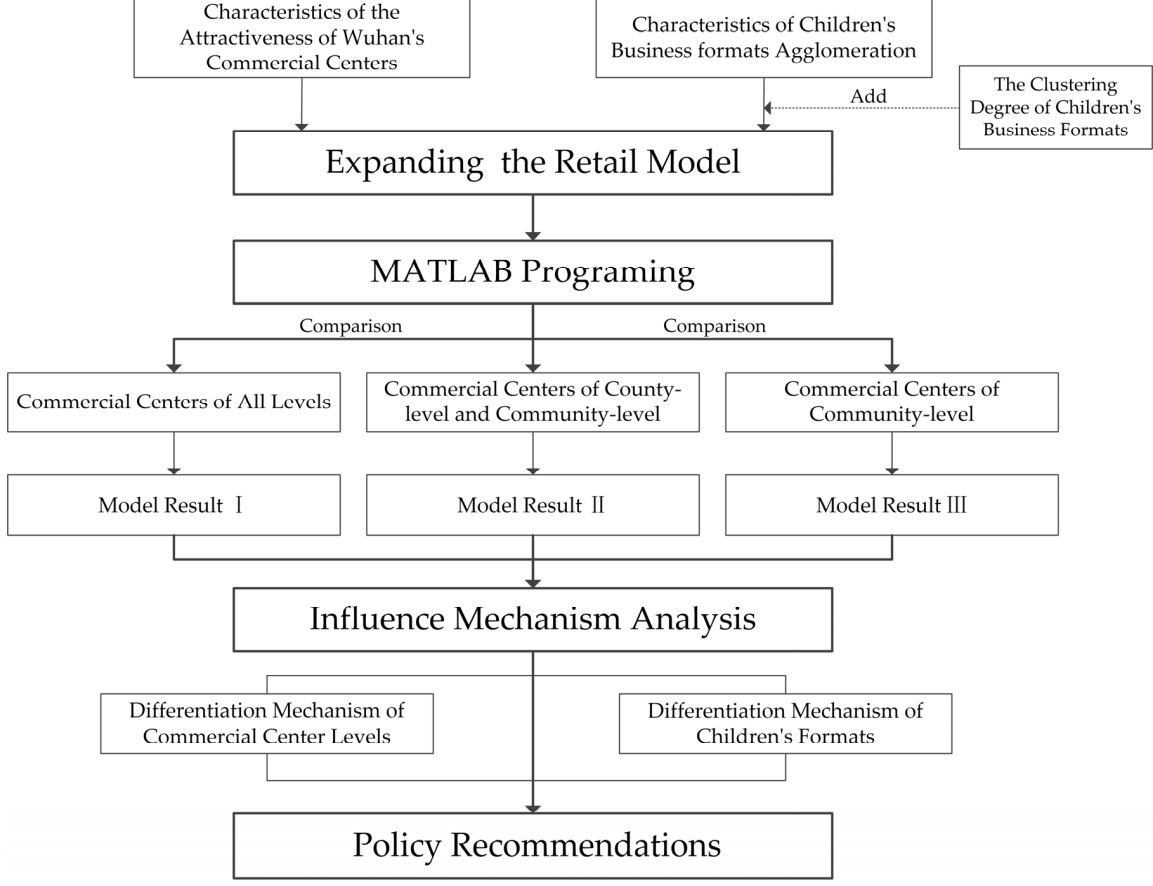

**Figure 1.** Analysis framework.

## 3. Materials and Methods

### 3.1. Research Material

#### 3.1.1. Built Environment Data for any Commercial Center

The data mainly include the name, grade, building outline, and geographical location of each commercial center. Among them, the name and level of commercial centers are mainly derived from the relevant content in the Wuhan Commercial Network Layout Plan (2016–2020), the Wuhan City Master Plan (2010–2020), and Shan et al.'s classification of Wuhan's commercial centers [36]. The building outline and geographic location were obtained from the AutoNavi Map API (2021). Based on these data, we defined 66 commercial centers in the main urban area of Wuhan, including six municipal-level commercial centers, 20 county-level commercial centers, and 40 community-level commercial centers.

#### 3.1.2. Children's Stores Data in Any Commercial Center

The research mainly collected two types of data: (1) data on the categories of children's stores and (2) data on the number of children's stores in commercial centers. The children's stores mentioned in this article refer to commercial operation sites that primarily target children, including children's retail stores, education stores, entertainment stores, and other stores. Children's entertainment stores include not only outdoor and indoor children's entertainment stores with fixed storefronts but also semi-mobile operating facilities for children's games that use public space in commercial centers to operate for an extensive period of time. Children's stores, as defined by the research, do not include non-profit public service facilities for children (e.g., mother and baby rooms, children's rest areas, etc.); comprehensive stores that do not have child-specific goals; stores that do not operate normally for one month or more or operate normally for less than one month without follow-up business intentions; small, fully self-service vending machines; temporary mobile stores; etc.

We conducted a three-month field survey of 66 commercial centers in the main urban area of Wuhan in 2021. According to non-participatory observations of the attributes and behaviors of visiting customers and store operators, 3000 children's consumer stores that met the research needs were selected, with the number of children's stores in each commercial center and each format counted separately. To minimize the deviation of results caused by individual differences between children's stores (e.g., differences in store area), we will increase or decrease the number of stores as appropriate in the counting process according to the actual situation of each store.

#### 3.1.3. Geographic Information Data of Wuhan

We collected administrative division data and traffic time consumption data for Wuhan from AutoNavi Map API (2021). The administrative division data included the main urban area of Wuhan and the administrative boundaries of each street. The traffic time data are the average minimum traffic time during the data collection period (31 August 2021 to 13 September 2021, working days 1700 to 2200, and rest days 1000 to 2200) for the four modes of transportation: walking, driving, cycling, and public transportation (including transfer time). After obtaining the minimum traffic time from any street to any commercial center, the average time was taken as the traffic time of any commercial center.

#### 3.1.4. LBS Location Service Data

The LBS location service dataset used in this study is user anonymous geographic location data obtained from when users use multiple apps (WeChat, Dianping, Meituan, etc.) in which they authorize the use of their location. The data include the anonymous ID of the user, the latitude and longitude of the user's residence, and the code for and time when the user visits the business center. The LBS data were collected from 66 commercial centers from 31 August 2021 to 13 September 2021 (holidays and periods of rain and snow were avoided), from 1700 to 2200 on weekdays and from 1000 to 2200 on weekends. After denoising, the invalid collection points for those passing by the commercial center, staying

on the water or on the road, and living and working in the commercial center, 7128 pieces of valid data were finally used.

*3.2. Research Methods*

3.2.1. Calculating the Clustering Degree of Children's Business Formats

The business point density index is widely used in the empirical analysis of business agglomeration. This study uses the density of children's stores (including the density of children's retail entertainment stores, education stores, other stores, and the overall density of children's stores) as an indicator of the children's format clustering degree. The density of children's stores is defined as the number of children's consumer stores in any commercial center or base area of the building outline. Ultimately, 264 pieces of valid data on the density of children's stores were obtained.

3.2.2. Measuring the Attractiveness of Commercial Centers

In an empirical analysis, the concept of customer attraction in commercial centers is often based on the interaction between customers and commercial spaces. In the case that a customer visits the same commercial center multiple times in a single day, only one spatial interaction was counted. We captured mobile communication devices within the range of 66 commercial centers using LBS data and used device IDs to identify the coordinates data of each consumer's residence. We used 108 sample blocks as the statistical unit, sequentially aggregated all street IDs to 66 commercial centers and the number of people interacting with each commercial center, and established a data matrix for the number of interactions between the sample blocks and commercial center spaces. Furthermore, we measure the attraction of any commercial center to any street based on Equation (2) and add the value of its attraction to all streets to obtain the attractiveness of any commercial center (Table 1).

**Table 1.** Matrix of spatial interaction among streets and commercial centers.

| Street | NO.4 | NO.1 | NO.30 | NO.39 | NO.35 | ...... |
|---|---|---|---|---|---|---|
| Baishazhou Street | 0.0057592 | 0.00280938 | 0.07978649 | 0.011658941 | 0.045933418 | |
| Baibuting Street | 0.0022283 | 0.00119986 | 0.00308536 | 0.016112444 | 0.014226945 | |
| Baofeng Street | 0.0028648 | 0.0026738 | 0.00305577 | 0.026355997 | 0.018716578 | |
| Beihu Street | 0.0040866 | 0.00122599 | 0.00367797 | 0.025337148 | 0.018798529 | |
| Caidian Street | 0.0051118 | 0.00340788 | 0.01299255 | 0.032800852 | 0.027050053 | |
| ...... | | | | | | |
| The value of Attractiveness | 1.1673702 | 1.04127612 | 1.66115807 | 3.826869759 | 3.594317598 | ...... |

Derived from LBS data.

3.2.3. Expanding and Solving the Retail Model Using MATLAB

With comprehensive consideration given to the research needs and data collection types, we finally selected the retail gravity model of physical and statistical mechanics for adjustment. The mathematical model is as follows:

$$T_{ij} = \frac{e_i p_i W_j^\alpha exp(-\beta C_{ij})}{\sum_k W_k^\alpha exp(-\beta C_{ik})} \tag{1}$$

where $T_{ij}$ represents the attractiveness of commercial center $j$ to residence $i$ (usually expressed by the number of customers and the probability that a customer at $i$ will choose business center $j$). $e_i$ is the per capita shopping demand, $p_i$ is the size of the population in residence $i$, $W$ is the construction area of any commercial center, $C$ is the comprehensive cost from the residence to the commercial center (expressed by distance or time consumption), and $k$ is the total number of competing commercial centers. $\alpha$ and $\beta$ are the adjustment coefficients for $W$ and $C$, respectively. In addition, we considered the overall shopping demand of residence $i$ as $Y_i = e_i p_i$, which is usually defined as $\sum_j T_{ij} = Y_i$.

Equation (1) can be regarded as a fast dynamic model based on the entropy maximization process to describe the flow of commercial centers [56]. In the process of competing for limited resources, various commercial centers achieve a dynamic balance of commercial scale and traffic resistance, forming a relatively stable retail distribution space pattern. The number and spatial distribution of commercial centers in this equilibrium state strongly depend on commercial scale parameter $\alpha$, distance cost parameter $\beta$, and the value of the input data. Any small change in the input data would yield a rapid reconfiguration to a new attractor state. In the process of expanding the classic model, scholars have proved that $\alpha$ is greater than 0, the value converges toward 1, and the value of $\beta$ is discrete but always greater than 0.

We made the following adjustments to the classic retail model.

(i) We used the number of visitors to define $T_{ij}$ and $Y_i$, so it can more accurately reflect the multi-frequency and multi-destination characteristics of modern consumer behavior. Therefore, we redefined $T_{ij}$ as the number of visitors from residence (street) $i$ to business center $j$ and $Y_i$ as the total number of visitors from residence (street) $i$ to all commercial centers. Finally, based on the definition of the spatial interaction matrix, we redefined the attractiveness of commercial center $j$ to residence (street) $i$ as

$$A_{ij} = \frac{T_{ij}}{Y_i} \tag{2}$$

(ii) In recent years, more and more studies have confirmed that the clustering degree of children's business formats has a potential impact on customer attraction, but this is rarely covered in the study of classic retail models. Therefore, we added the clustering degree of children's formats $Z_j$ to meet the research needs, and expanded the term $exp(-\beta C_{ij})$ to $exp(aC_{ij} + bZ_j)$.

$$Z_j = \sum_n F_n \ (n = 1, 2, 3, 4) \tag{3}$$

(iii) The traffic cost has a stable negative impact on the attractiveness of commercial centers, while the impact of the clustering degree of children's business formats is relatively positive, so we assume $\alpha < 0$, $\beta > 0$. Besides, Since the value of $\alpha$ always converges to 1, we defined $\alpha = 1$.

The improved model expression is as follows:

$$A_{ij} = \frac{W_j exp(aC_{ij} + b\sum_n F_n)}{\sum_k W_k exp(aC_{ik} + b\sum_n F_n)} \tag{4}$$

In Equation (4), we defined residence (street) $i$ as 108 sample blocks and $j$ as 66 commercial centers. $\sum_n F_n$ was used to express the total clustering degree of the children's business formats, and $C$ was used to express the traffic cost.

To solve the influence mechanism of the density of various children's consumer service stores on commercial centers at various levels, we used MATLAB to compile the calculation program. First, three groups of controlled trials were established according to the classification of commercial centers: full-level commercial centers (66), county-level and community-level commercial centers (60), and community-level commercial centers (40). Next, the sample street code (108 rows $\times$ 2 columns) and the total number of visitors to the sample street (108 rows $\times$ 1 column) were determined. Thereafter, three control group datasets were input: the spatial interaction probability matrix (108 rows $\times$ 66 columns/108 rows $\times$ 60 columns/108 rows $\times$ 40 columns), the spatial interaction matrix (108 rows $\times$ 66 columns/108 rows $\times$ 60 columns)/108 rows $\times$ 40 columns), the business center-street traffic time-consuming matrix (108 rows $\times$ 66 columns/ 108 rows $\times$ 60 columns/108 rows $\times$ 40 columns), and the business center attributes (66 rows $\times$ 7 columns/60 rows $\times$ 7 columns/40 rows $\times$ 7 columns). By changing the independent variables, we could compare the model output results to determine the optimal model, which was selected based on reliability, T-stat significance, the principle of

maximum R$^2$, and minimum error. Finally, the correlation coefficients of each influencing factor were derived.

## 4. Results and Analysis

*4.1. Characteristics of Children's Business Agglomeration in Wuhan's Commercial Centers*

### 4.1.1. Active Children's Business Formats

First, among the 66 sample commercial centers we surveyed, there were more than 3000 children's stores in total. Additionally, more than 90% of the commercial centers in the main urban area had functions associated with children's consumption and related services (Figure 2). Second, among the 66 sample business centers, approximately one-third of the business formats were children's formats (33.8%), which is nearly three times the average level in China in 2020 (12.5%, according to the "2021 China Winning Business Big Data Report"). Third, the density of children's stores was clustered at 0–0.5 per hectare (Figure 3), with an average clustering degree of 0.65 per hectare and a maximum degree of 2.98 per hectare (Figure 4). Therefore, it can be seen that children's businesses in Wuhan's commercial center are generally active.

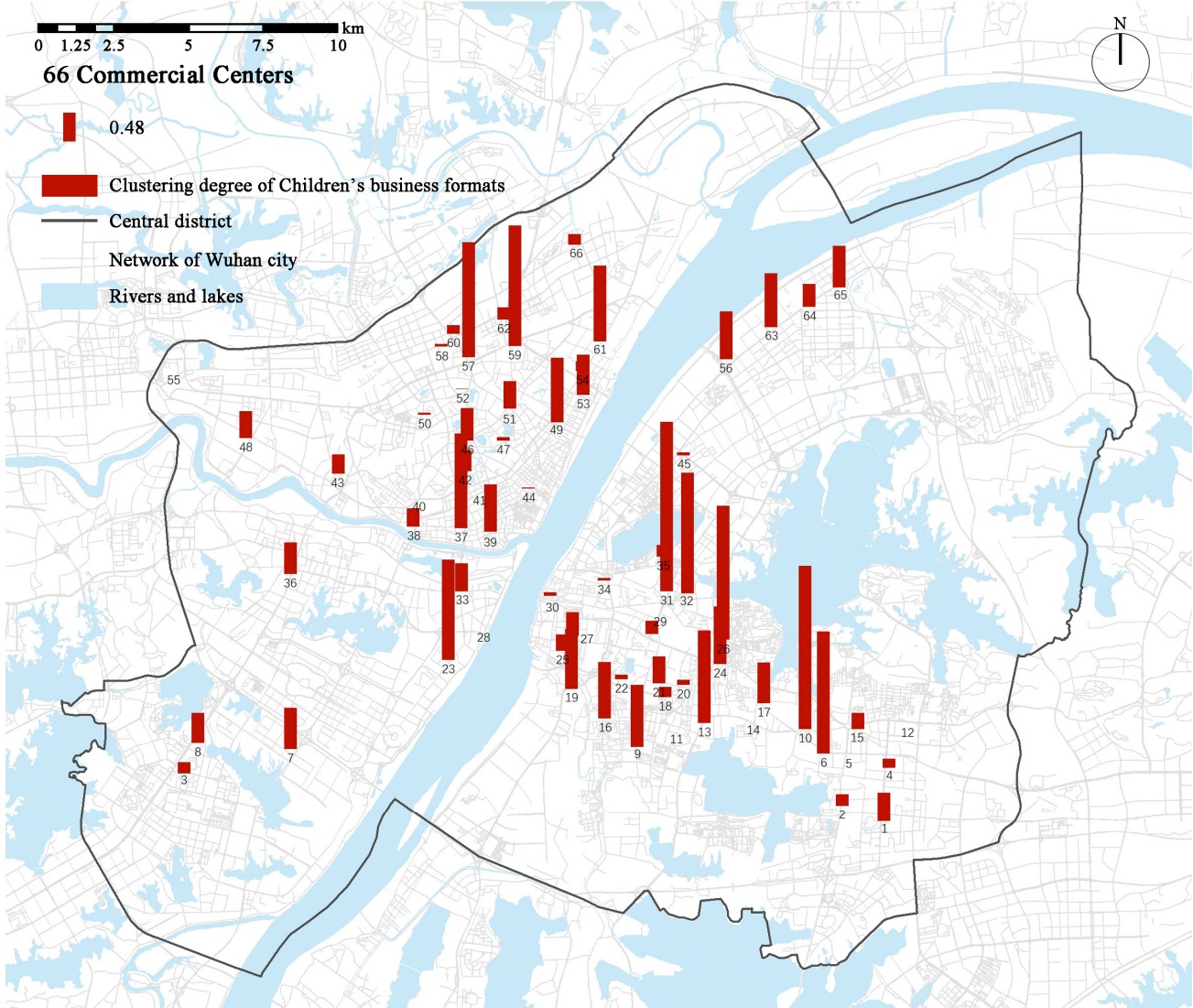

**Figure 2.** Ninety percent of the sample commercial centers have different degrees of children's business distribution. Data sources: the field investigation data.

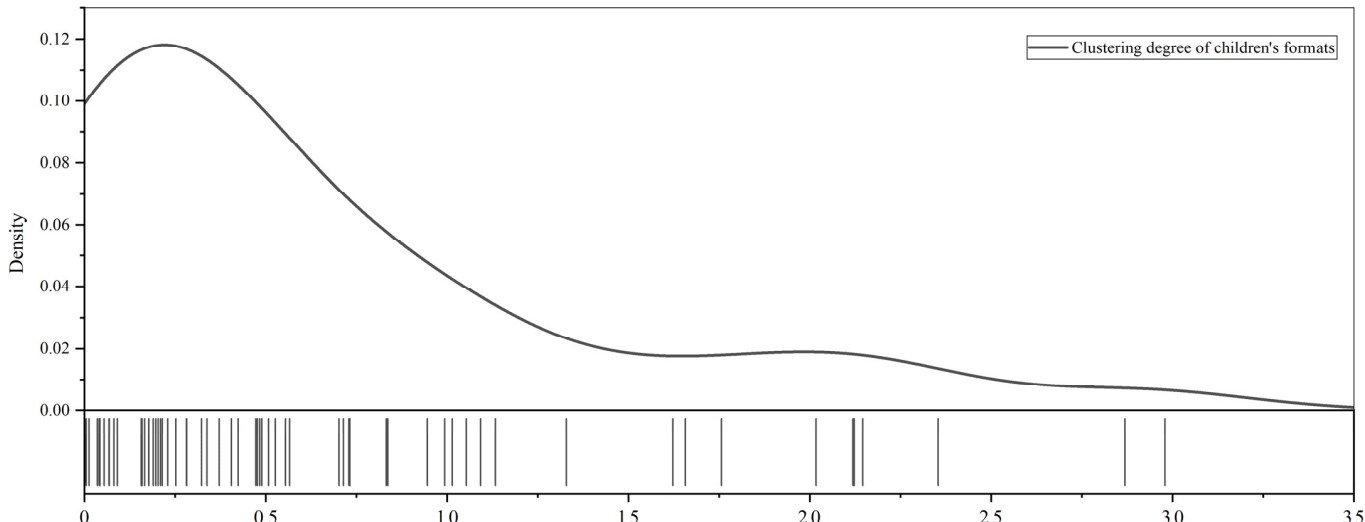

**Figure 3.** Kernel density estimation of clustering degree of children's business format in Wuhan's commercial centers. Data sources: the field investigation data.

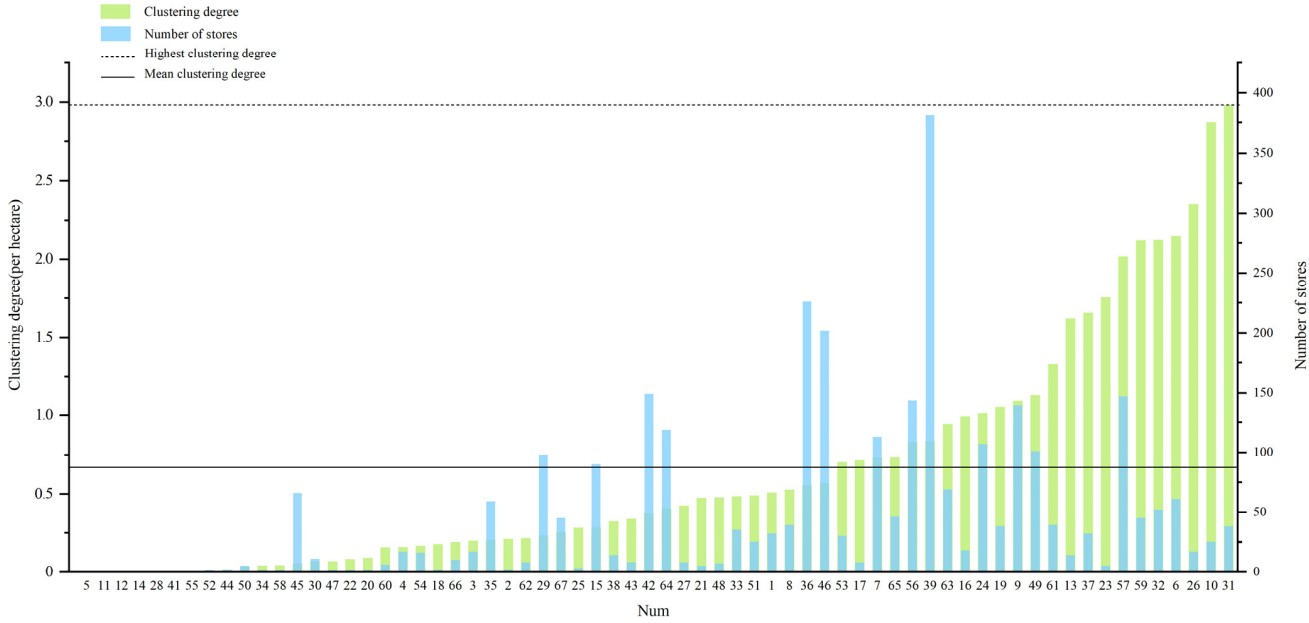

**Figure 4.** The clustering degree of children's business formats and the number of stores in the sample business centers. Data sources: the field investigation data.

### 4.1.2. Unbalanced Layout of Children's Formats in Different Commercial Centers

First, we find that the standard deviation of the overall clustering degree of children's stores in 66 commercial centers reached 0.74 per hectare. Figure 5 shows that the lower the level of commercial centers, the greater the gap between the median and the mean of the data group and the stronger the imbalance in children's business format layout.

Second, the average clustering degree of children's retail, entertainment, and education stores was 0.18, 0.12, and 0.31 per hectare, respectively. The layout and investment of various children's formats differed significantly—there was a greater number of children's education stores, and its distribution was more extensive.

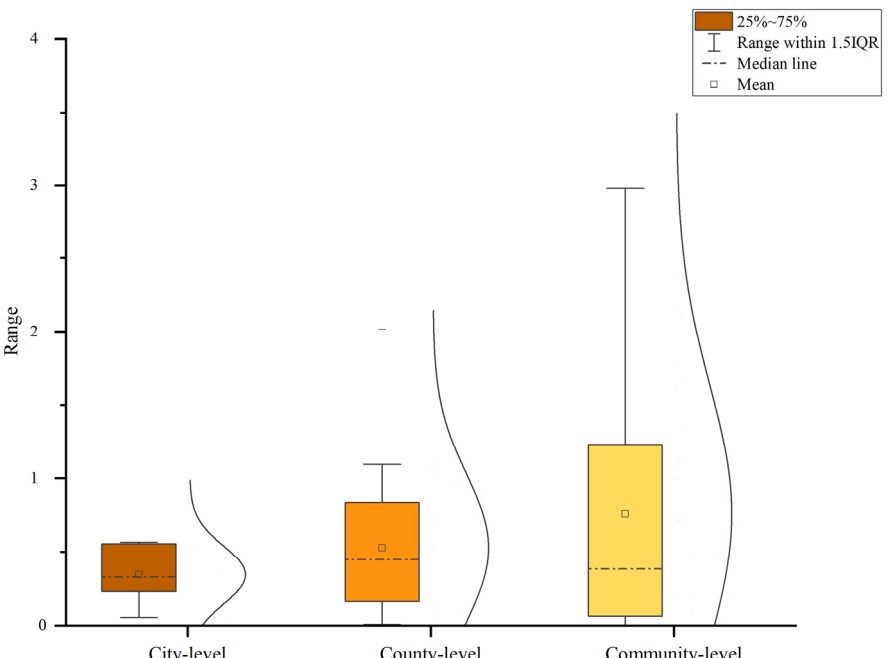

**Figure 5.** Unbalanced distribution of children's formats in Wuhan's commercial centers of different levels. Data sources: the field investigation data.

Furthermore, as shown in Figure 6, the children's business formats with the largest differences were children's education and children's entertainment, with a long "tail", an obvious gap between the high-value and low-value end, and a prominent "single-peak non-equilibrium" structure. The difference in the layout of children's retail formats was relatively small, showing a pattern orientation of "a primary and secondary multi-peak and non-equilibrium". The uneven distribution of various types of children's businesses in 66 commercial centers varied significantly.

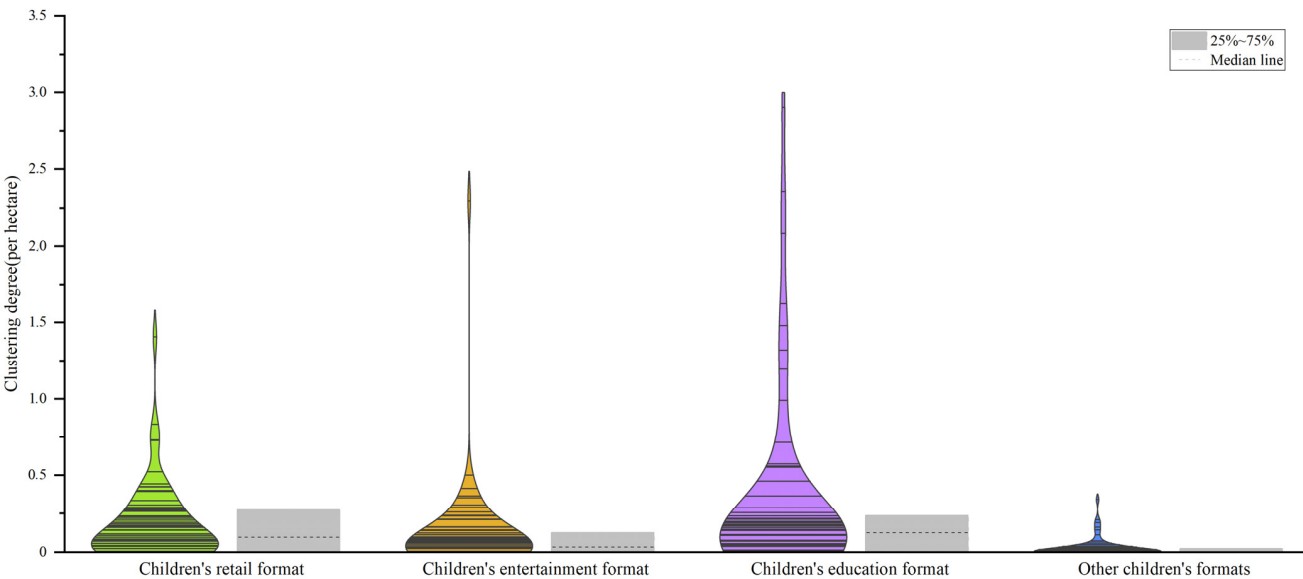

**Figure 6.** Violin diagram of the clustering degree of various children's business formats in 66 commercial centers in Wuhan. Data sources: the field investigation data.

Finally, as shown in Figure 7, in the city- and district-level commercial centers, more than half of the children's business formats were children's retail (accounting for 60.4% and 52.7%, respectively). Their average clustering degrees were 0.18 per hectare and 0.12 per

hectare, respectively. The children's education business format ranked second (accounting for 29.4% and 24.6%, respectively), and the average clustering degrees were 0.10 per hectare and 0.16 per hectare, respectively. The children's entertainment format occupied a larger proportion (55.8%) of stores found in community-level commercial centers, with a high average clustering degree of 0.45 per hectare, followed by the children's retail format (21.6%), with an average clustering degree of 0.14 per hectare. Therefore, there are differences in the proportions of various types of children's businesses at different levels of commercial centers.

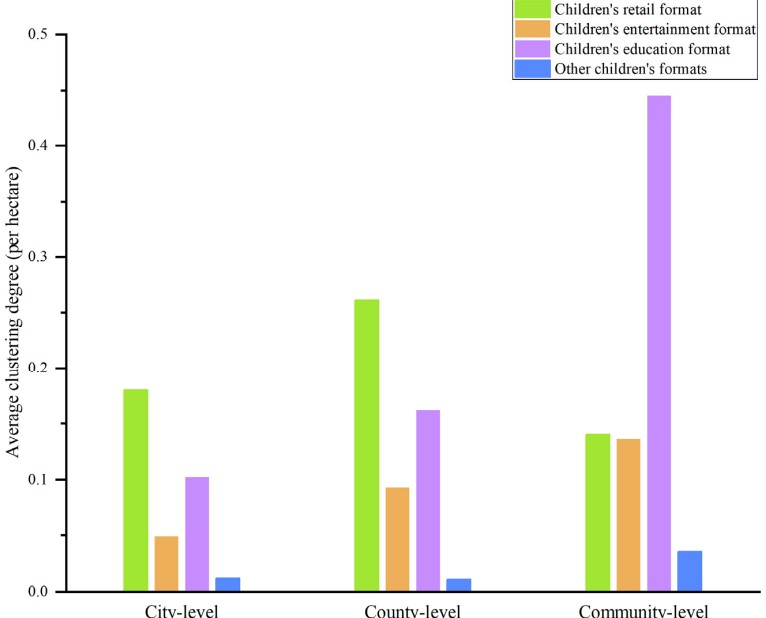

**Figure 7.** The difference in the proportion of 4 types of children's business formats in 66 commercial centers in Wuhan. Data sources: the field investigation data.

### 4.2. Characteristics of the Attractiveness of Wuhan's Commercial Centers

#### 4.2.1. Gradient Characteristics of Commercial Center's Attractiveness

Based on the measurements of the commercial centers, we created classification statistics on attractiveness. As shown in Figure 8, the attraction force of nearly 90% of the commercial centers shows a gradient characteristic that increases with level. The average values of customer attraction of city-, county-, and community-level commercial centers were 7.7, 2.0, and 0.5, respectively, and there was an obvious gradient difference in the adsorption capacity of commercial centers at each level.

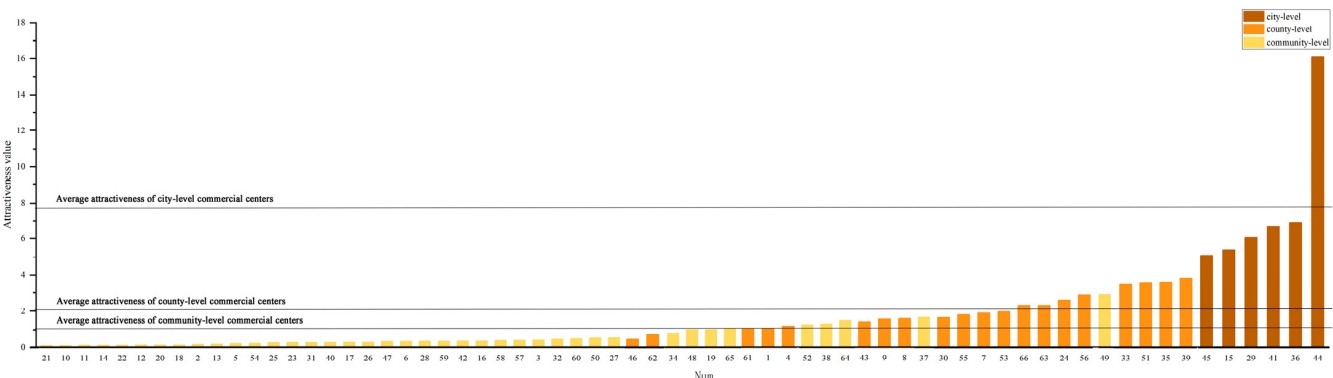

**Figure 8.** Gradients and intra-level differences in attractiveness of Wuhan's commercial centers. Data sources: derived from LBS data (Table 1).

### 4.2.2. Difference in the Attractiveness of Commercial Centers of the Same Level

Comparing the difference in customer attraction between same-level commercial centers (Figure 8), we found the following. (1) In city-level commercial centers, while most commercial centers tended to have similar levels of customer attraction, the attractiveness of the No. 45 commercial center was slightly lower than that of other commercial centers at the same level, and that of No. 44 was significantly higher than that of other commercial centers. (2) In the county-level commercial centers, the attractiveness of the No. 46 commercial center was lower than that of other commercial centers of the same level, while that of No. 39 was slightly higher than the other commercial centers. (3) In community-level commercial centers, the attractiveness of most commercial centers was low, with No. 49 being significantly higher than the average for its class.

### 4.2.3. Non-Linear Relationship between the Attractiveness of Commercial Centers and the Clustering of Children's Business Formats

We used the number of commercial centers as the horizontal axis and the standardized value as the vertical axis to draw a line graph of the standardized value of each variable in order to observe its relationship with the attractiveness value (Figure 9). The analysis showed that the relationships between attractiveness and most factors do not conform to the general linear relationship; thus, it is misleading to analyze the relationship between the agglomeration degree of children's business formats and the attractiveness of commercial centers. To describe the non-linear relationship between the variables, it is therefore necessary to study and improve the retail gravity model.

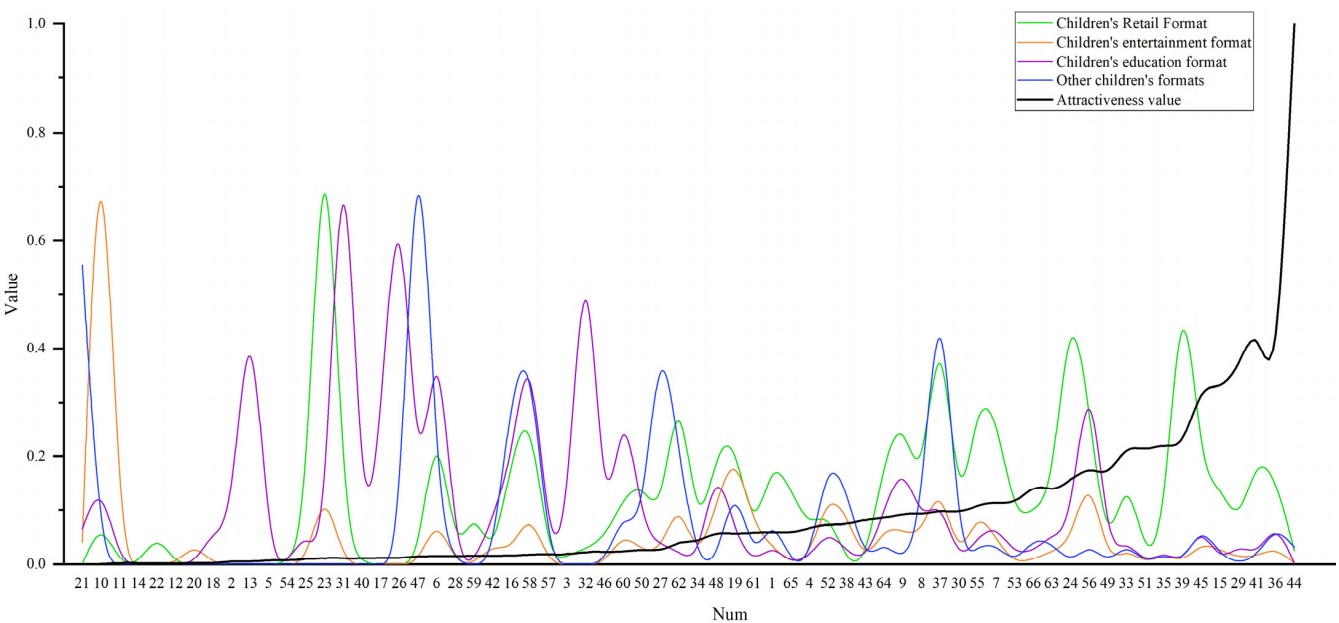

**Figure 9.** Line chart about the relationship between attractiveness and the clustering degree of children's formats. Data sources: the field investigation data, AutoNavi Map API data, and LBS data.

### 4.3. Influence Mechanism of the Clustering Degree of Children's Business Formats on the Attractiveness of Commercial Centers

#### 4.3.1. Modeling Results

In this study, we constructed a retail gravity model by using the interaction probability between commercial centers and township streets as the explained variables and defining the traffic time and agglomeration of children's business forms as the explanatory variables. The clustering degree of children's business formats includes the clustering degree of the children's retail, entertainment, education, and other formats. We used SPSS to test the factors and found no obvious correlations between them (Table 2). Due to the limited number of samples investigated, we completed the modeling and solving by setting up

three control groups—full-level commercial centers, county-level and community-level commercial centers, and community-level commercial centers—to robustly explore the effects of various explanatory variables on different-level commercial centers (Tables 3–5).

**Table 2.** Correlation matrix of explanatory variables.

|  | Children's Retail Format | Children's Entertainment Format | Children's Education Format | Other Children's Formats | Traffic Time |
|---|---|---|---|---|---|
| Children's retail format | 1 | 0.203 | −0.008 | 0.099 | −0.023 |
| Children's entertainment format | 0.203 | 1 | 0.056 | 0.052 | 0.089 |
| Children's education format | −0.008 | 0.056 | 1 | 0.013 | 0.048 |
| Other children's formats | 0.099 | 0.052 | 0.013 | 1 | −0.189 |
| Traffic time | −0.023 | 0.089 | 0.048 | −0.189 | 1 |

Data sources: the field investigation data, AutoNavi Map API data, and LBS data.

**Table 3.** Modeling results for commercial centers of all levels.

| Explanatory Variables | Influence Level | | Model Estimation | | Observed Value vs. Calculated Value |
|---|---|---|---|---|---|
|  | Correlation Coefficient | T-Stat | $R^2$ | Reliability |  |
| Traffic time | −0.031308 | −844.94 |  |  |  |
| Children's retail format | −0.0087432 | −15.149 |  |  |  |
| Children's entertainment format | −0.0080871 | −10.452 | 0.611 | 0.0289 | y = 0.9171x + 0.001256 |
| Children's education format | 0.019973 | 63.941 |  |  |  |
| Other children's formats | 0.16289 | 45.68 |  |  |  |

Data sources: the field investigation data, AutoNavi Map API data, and LBS data.

**Table 4.** Modeling results for commercial centers of county-level and community-level.

| Explanatory Variables | Influence Level | | Model Estimation | | Observed Value vs. Calculated Value |
|---|---|---|---|---|---|
|  | Correlation Coefficient | T-Stat | $R^2$ | Reliability |  |
| Traffic time | −0.033497 | −753.76 |  |  |  |
| Children's retail format | −0.018948 | −29.548 |  |  |  |
| Children's entertainment format | −0.003684 | −4.3877 | 0.446 | 0.0236 | y = 0.5352x + 0.000609 |
| Children's education format | 0.016845 | 49.463 |  |  |  |
| Other children's formats | 0.16441 | 40.411 |  |  |  |

Data sources: the field investigation data, AutoNavi Map API data, and LBS data.

**Table 5.** Modeling results for commercial centers of community-level.

| Explanatory Variables | Influence Level | | Model Estimation | | Observed Value vs. Calculated Value |
|---|---|---|---|---|---|
|  | Correlation Coefficient | T-Stat | $R^2$ | Reliability |  |
| Traffic time | −0.039562 | −397.37 |  |  |  |
| Children's retail format | 0.01977 | 11.184 |  |  |  |
| Children's entertainment format | −0.036233 | −30.036 | 0.309 | 0.0167 | y = 0.1766x + 0.000425 |
| Children's education format | −0.0018477 | −4.2597 |  |  |  |
| Other children's formats | 0.23506 | 47.86 |  |  |  |

Data sources: the field investigation data, AutoNavi Map API data, and LBS data.

### 4.3.2. The Impact of Children's Retail Format

Children's retail format had a limited influence on the attractiveness of commercial centers. By observing the relationship between the two values (Tables 1 and 2), we can see that this type of business format had a weak and unstable effect on the attraction of city-level commercial centers. By comparing Tables 2 and 3, we find that the correlation coefficient of children's department stores changed significantly from negative to positive, indicating that children's department stores have an inhibitory effect on the attraction of county-level commercial centers. The correlation coefficient of children's retail format in Table 3 is 0.01977, indicating that it has a strong appeal to consumers in community-level commercial centers. On the one hand, this may be because children themselves have no spending power, and retail commodities such as children's food, toys, and clothing are not necessities of life for all consumers. Therefore, for more consumers, other types of stores in city- and district-level commercial centers are more attractive. However, the acquisition threshold of such products is low and the space demand for store operations is small. Community-level commercial centers can already meet the needs of most consumers to obtain children's general merchandise. At the same time, with the popularity of online shopping, consumers' expectations of the shopping travel costs for children's daily necessities have decreased. More consumers consider that they do not need to go to high-level commercial centers to buy children's daily necessities. In addition, this phenomenon is closely related to the retail format characteristics. This type of format cannot use the characteristics of "accompanying reading" and "accompanying playing" to further expand the customer attraction effect, such as entertainment and education formats, so it is relatively passive in promoting the role of attracting customers.

### 4.3.3. The Impact of Children's Entertainment Format

As shown in Table 3, the correlation coefficient of children's entertainment format on the attraction of community-level commercial centers was $-0.036233$, indicating that it has a significant inhibitory effect. A comparison of Tables 1 and 2 showed that the absolute value of the correlation coefficient of this format and its change are both small, and the effect at the city-level commercial center is not significant. Several reasons explain this phenomenon. First, children's entertainment venues are not limited to commercial centers. Community centers, city parks, and other indoor or outdoor public spaces can also provide playgrounds for children. "Pay to play" has become the norm in major Chinese cities, but Severcan's research found that children prefer free public spaces. This makes children's entertainment facilities in commercial centers not the only or optimal choice for consumers. Therefore, this type of business has a very limited influence on the attraction of commercial centers in general. Second, the scope of design allowed for children's play places in community-level commercial centers was limited, the budget was small, there was only one set of amusement facilities, and there was insufficient daily management and maintenance. Merely placing rudimentary equipment and facilities is not enough to attract children, nor can it meet the needs of most modern consumers. Therefore, the high concentration of children's entertainment formats has inhibited improvement in the attractiveness of community-level commercial centers. Third, city-level commercial centers have more advantages in terms of the novelty and diversity of children's amusement facilities. While there are many types of businesses in these commercial centers, it should be noted that the phenomenon of "an inch of land and an inch of gold" directly conflicts with the large space demand of children's entertainment stores, resulting in relatively poor competitiveness in terms of revenue per store. Consumers in city-level commercial centers generally have diverse visiting purposes; therefore, time spent in a single store is usually limited. However, children often need to stay in such stores for a long time, which leads to single-purpose consumption travel that can better meet the needs of children's play time. That said, in the case that a consumer trip has multiple purposes and cannot provide sufficient time to stay in a children's entertainment store, most consumers often choose to give up visiting such stores.

### 4.3.4. The Impact of Children's Education Format

A comparison of Tables 1 and 2 revealed that the correlation coefficients of children's education formats are all positive, and the absolute value is decreasing, thereby indicating that this type of format has a promoting effect on city-level customer attraction. In Tables 2 and 3, the correlation coefficient of this type of business changed from a positive to a negative value, demonstrating that the children's education format also has a promoting effect on attracting customers in county-level commercial centers. The correlation coefficient of this type of business in Table 3 was −0.0018477. This small value indicates a weak role for community-level commercial centers. First of all, the "accompanying reading" model generated by education and training stores can bring a large number of consumers and increase the length of consumers' stay. It can also effectively stimulate the potential consumption power of parents, thereby amplifying the "siphon aspect" of high-level commercial centers. At the same time, education and training services possess the characteristics of continuity and periodicity and require consumers to maintain a long-term relationship with service personnel, which provides a long-term and stable passenger flow for the commercial centers. Compared with children's retail stores, such stores are more dependent on the offline physical environment and have strong site adaptability, which makes them more resilient to the impact of online shopping. Second, children's education stores in Wuhan have formed a degree of differentiated division of labor in commercial centers of different levels. The city-level commercial centers have focused on new types of children's quality education and art education, such as sensory integration training, educational thinking enlightenment, and emotional social and language communication guidance. In contrast, county-level commercial centers have emphasized traditional quality education, e.g., instrumental music, art, and dance, and subject education, e.g., Chinese, mathematics, and English. Some stores also provide childcare services. This division of labor is in line with customers' consumption travel expectations, thus enabling children's education stores to be developed in both city- and county-level commercial centers. However, for community-level commercial centers, the high cost, venue, and talent threshold restrictions of such stores make it difficult to adapt to consumer needs.

### 4.3.5. The Impact of Traffic Resistance

Traffic impedance showed an obvious and stable inhibitory effect on commercial centers at all levels. The accessibility of commercial centers significantly affects consumers' willingness to travel. Moreover, the higher the cost of transportation, the lower the consumers' willingness to shop. Comparatively speaking, consumers have a higher acceptance threshold for shopping travel costs in high-grade commercial centers; therefore, the inhibitory effect of traffic impedance decreases slightly as the commercial center level increases.

### 4.3.6. Differences in the Influence between Factors

In general, the effects of different factors on different levels of commercial centers vary. Children's retail format has a strong inhibitory and promotional effect on county- and community-level commercial centers but has no significant effect on city-level commercial centers. The children's entertainment format has a relatively large inhibitory effect on the attraction of community-level commercial centers but has a weaker impact on city-level commercial centers. Children's education format has a significant promoting effect on the attractiveness of county-level commercial centers and also appears to have a promoting effect on city-level commercial centers, but has a weak impact on community-level commercial centers. Traffic impedance showed a stable and strong inhibitory effect at all levels of commercial centers.

## 5. Conclusions

### 5.1. Summary

This study investigated the mechanism of the clustering degree of children's business formats on the attractiveness of commercial centers and examined the similarities and

differences between the effects of the mechanism on different levels of commercial centers. The results of this study are as follows. (1) First, children's business in Wuhan's commercial centers is generally active, and different types of children's business formats have an unbalanced layout at different levels of business centers. (2) Second, there is not only a grade gradient difference in the attraction of commercial centers in Wuhan but also a significant gap in the attractiveness of commercial centers at the same level. (3) Various types of children's business formats show different directions and levels of influence on different levels of commercial centers. For city-level commercial centers, the preference for children's education and entertainment formats, to some extent, plays a role in promoting them; for county-level commercial centers, children's education format is the most important, while too much investment in children's retail format may not meet the expectation of the marketers; and for community-level commercial centers, investment in children's retail format shows the best results. (4) Traffic resistance shows a steady inhibitory effect and is more sensitive in low-grade commercial centers than in high-grade commercial centers.

This study innovatively optimizes the retail gravity model. Utilizing the data of first-tier cities in China to test the validity of the model, it enriches the theoretical research on the retail gravity model and provides a new understanding of the influence mechanism of commercial center attractiveness. Further, we explored the relationship between the clustering degree of children's business formats and the attractiveness of commercial centers and clarified the differences in the impact mechanism at different levels of commercial centers.

*5.2. Policy Recommendations*

The risks to the current children's businesses' prosperity are obvious. These include the impact of the COVID-19 epidemic, the "double reduction" (reducing the homework burden of students in compulsory education and the burden of off-campus training) policies of the government, increased consumer demand, and regional economic decline. The chain problems these issues cause cannot be underestimated. Therefore, to fully understand the potential of children's consumption, it is critical to control the proportion of business formats and coordinate the relationships between the distribution of children's business formats and various commercial centers. This study puts forward suggestions on the configuration of children's business formats in commercial centers from the following five perspectives:

(1)     The revitalization of urban high-level commercial centers in the global post-epidemic era

This study found that investment in children's education formats can effectively improve the attractiveness of urban high-level commercial centers. First, major international cities (London, New York, Hong Kong, etc.) have entered the post-epidemic era, wherein the recovery of urban economies and commercial centers after the epidemic is particularly urgent. Second, families' willingness to invest in children's education continues to increase, and the children's education consumer market is still expanding, which remains a global development trend. Third, experiential business formats (e.g., education, entertainment, etc.) will be hit harder during the epidemic, and these kinds of formats will also rebound faster after the epidemic. Therefore, in the post-epidemic era, integrating children's education formats in high-level commercial centers is an effective means of promoting their vitality.

(2)     The strengthening of children's retail formats in commercial centers under the "government-driven" adjustment of market failure

On the one hand, this study found that investment in children's retail formats is extremely effective in community-level commercial centers, while children's entertainment format has a negative impact on their appeal. On the other, the degree of residents' reliance on community business services has gradually increased and their growing preference for retail shopping in community centers and nearby living circles is becoming increasingly obvious. Nowadays, the children's business model of "pay to play" is sweeping the world, and low-level commercial centers, such as community-level commercial centers that meet the daily lives of residents as their main functions, have also followed suit. The children's

retail format in Wuhan's community-level commercial centers accounts for only one-fifth of the total children's businesses at present, lagging far behind the children's entertainment business (55.8%). This makes the configuration of the children's business format misaligned with the needs of residents, thus complicating further investment to meet consumer expectations. Therefore, to benefit users and meet the shopping needs of nearby residents, market capital should be guided to invest in the children's department store format in community-level commercial centers, and the government's public policies should be reviewed and fulfilled. In addition, targeted support should be provided to communities in which the supply and demand of children's retail formats are not well aligned.

(3)   Transformation of children's education format in commercial centers in response to China's "double reduction" policy

Although this study confirmed the driving effect of children's education format on the attraction of commercial centers, subject and traditional quality education facilities will inevitably decline under the objective constraints of the Chinese government's implementation of the "double reduction" policy. This will not only limit children's education stores but also affect the stores attached to them. Therefore, the composition of the existing education format should be adjusted by cultivating a new type of children's education format that can continue to play a positive role in the children's education format and make up for the negative reduction effect of the decline in traditional educational facilities on the attraction of commercial centers. In addition, relying on the advantages of relatively low store rents and customer travel costs in county-level commercial centers, another feasible optimization strategy would be to introduce disciplinary and non-disciplinary education and training institutions to provide after-school care services for children.

(4)   Flexible format adjustment to improve the mismatch between commercial center level and children's business formats

Research shows that there is an obvious mismatch between the configuration of children's business formats in Wuhan's commercial centers and their impact on customer attraction, which makes it difficult to successfully invest in children's business formats and achieve the desired results. In city-level commercial centers, the increase in children's education, entertainment, and other experiential business formats can be more appealing to consumers, but the current situation is that the dominant business format is children's retail stores. Similarly, investment in the children's education format in county-level commercial centers is the most effective, and it is more beneficial to increase the proportion of children's retail formats in community-level commercial centers. However, the status quo is that the children's retail format accounts for the highest proportion of stores found in district-level commercial centers, while the children's entertainment format accounts for a larger proportion in community-level commercial centers. Therefore, it is necessary to gradually adjust the configuration of children's business formats in commercial centers at all levels in Wuhan in flexible ways, such as business integration and business upgrading.

(5)   Targeted optimization of business centers to make up for the shortcomings of children's business formats

Based on the measurement of the attraction of 66 commercial centers in Wuhan, we found that the commercial centers of No. 45 (city level), No. 46 (county level), and No. 47 (community level) had the lowest attractiveness among others of the same grade. From the perspective of children's business configuration, this research attempts to put forward targeted optimization suggestions for these three commercial centers: (i) The proportion of children's retail, entertainment, and education in the No. 45 commercial center (city-level) accounts for 41.1%, 16.3%, and 38.1% of the total children's business, respectively, which is basically in line with the average proportion of children's business in commercial centers at the same level. However, the type of children's education format is too conservative—mainly music education, art education, etc. Therefore, we suggest that the proportion of children's business formats be appropriately adjusted with dependence

on children's education stores reduced, and the aggregation of new children's quality education stores prioritized. (ii) The overall situation of children's business formats in the No. 46 commercial center (county level) is weak, but it is a feasible optimization strategy to moderately increase the education-oriented children's business format. (iii) The children's business format of the No. 47 commercial center (community-level) is primarily children's education. The problem here concerns the mismatch between the children's business formats and the level of commercial centers. Thus, we recommend reducing dependence on children's education and supporting the supply of daily children's general merchandise in cooperation with community children's public service facilities.

## 6. Limitations and Future Research

As with any study, this study has certain limitations. First, owing to its data-driven and model research approach, the generalizability of the results is limited. Second, given the short duration of data collection and the limited scale of the collection sites, the model construction is limited. In addition, the data acquisition method has limitations, resulting in insufficient data accuracy and errors in the extraction of shopping and residential locations. Thus, the generality of this study needs to be further improved.

Future studies could investigate children's business formats in different cities to verify whether the patterns revealed in this study would work in diverse settings. In addition, consumer attributes and behavioral characteristics were not fully covered in this study. It would be interesting to further improve this research by combining subjective consumer surveys and interviews.

**Author Contributions:** Conceptualization, Z.S. and X.S.; methodology, Z.S.; software, Z.S. and X.S.; validation, Z.S., X.S. and M.Y.; formal analysis, Z.S. and X.S.; investigation, Z.S. and X.S.; resources, Z.S.; data curation, Z.S.; writing—original draft preparation, Z.S. and X.S.; writing—review and editing, Z.S. and X.S.; visualization, X.S.; supervision, Z.S.; project administration, Z.S.; funding acquisition, Z.S. All authors have read and agreed to the published version of the manuscript.

**Funding:** This research was funded by Wuhan Knowledge Innovation Special Dawning Project, grant number 20221569; the Fundamental Research Funds for the Central Universities, grant number 2021WKZDJC015; the National Natural Science Foundation of China, grant number 51708233.

**Institutional Review Board Statement:** Not applicable.

**Informed Consent Statement:** Not applicable.

**Data Availability Statement:** All open source data in this article are from: https://lbs.amap.com/, accessed on 19 July 2021.

**Conflicts of Interest:** The authors declare no conflict of interest.

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
