# Peer review of "Exploring the Relationship between the Clustering Degree of Children’s Business Formats and the Attractiveness of Commercial Centers in Wuhan by Modifying the Classic Retail Model"

_land, doi:10.3390/land11081175_

Round 1
Reviewer 1 Report
1. Please explain "community level". Does "traffic time" mean "peak-hour"?
2. For the attractiveness of commercial centers, how did the authors calculate the probability of interaction between the commercial center and consumer space? Probability or proportion?
3. Please provide correlation matrix for the explanatory variables. Are there highly-correlated variables?
4. Why considering 3 separate models? Can you consider the dummy variables to indicate "county-level" or "non-county-level" in one model? What is community-level...
5. Please explain the modeling results in a more professional/scientific way.
6. Some statements are note clear. For example, the lines 494-496 in page 14. What do you mean "talent threshold"?
Author Response
请参阅附件。

Reviewer 2 Report
Dear authors, the manuscript is presented well where detail about methods, data and analysis is clearly presented which broaden the current understanding of the topic. I congratulate the authors for producing good results. I shall highlight some inadequacies and the authors might improve the quality and readability of this research paper accordingly.
1.It is necessary to strengthen the analysis of the reference significance of Chinese cases to international cities and enhance the value of the paper.
2. Please, briefly add future perspectives and further applications of this specific research work.
3. Figure 1 is not clear.
Reviewer 3 Report
The theme of the article is interesting because, as specified in the introduction, "children play an increasingly active role in the consumer group, and the commercialization of children’s space has also become a new development trend on a global scale".
Precisely on the basis of this assumption, the article would also benefit from just a couple of references, to be added in the introduction, to case studies analyzed in other territorial contexts: although if not directly focused on the link between attractiveness if commercial centres and children's businesses formats agglomerations, other studies through a broader overview on the topic can help to better understand what is illustrated in the article.
At the same time, again in the introduction, the article would benefit from even a brief elaboration to the background, especially regarding the concept of scale applied to commerce and consumption (since commercial scale, business scale, consumer scale etc. are repeatedly mentioned).
From this point of view, the framework of bibliographic references could also be further improved by adding more references to the link between retail, consumption and identity for a specific group such as children.
As for English language and style finally, it is recommended only to make the minor changes required by rereading the text.
Reviewer 4 Report
Authors examine "Exploring the Relationship between the Clustering Degree of Children's Business Formats and the Attractiveness of Commercial Centers in Wuhan by Modifying the Classic Retail Model" seems an interesting work, and report 1-4 important observations mentioned in the abstract. Some suggestions:
1. Authors need to build strong conceptual based and theoretical support for the Retail model
2. Authors need to further add important studies on the different types of children's business model
3. Authors fail to link the Gravity model in assessing the relations between children's business and the attraction of commercial centers
Reviewer 5 Report
I highly appreciate the article for:
- interesting research topic;
- clearly defined research objective;
- clearly defined research process;
- recommendations contained in the final conclusions.
Changes that should be made in the article:
- improving the linguistic and punctuation quality of the entire text;
- improving the quality of all charts and figures so that they are legible and transparent;
- adding data sources under the figures and tables;
- development of all abbreviations used for the first time in the text, which may be incomprehensible to a foreign reader.
